# Impairment of Microcirculation Parameters in Patients with a History of Diabetic Foot Ulcers

**DOI:** 10.3390/medicina61010002

**Published:** 2024-12-24

**Authors:** Julien Vouillarmet, Audrey Josset-Lamaugarny, Myriam Moret, Christine Cugnet-Anceau, Paul Michon, Emmanuel Disse, Dominique Sigaudo-Roussel

**Affiliations:** 1Service d’Endocrinologie, Diabète et Nutrition, Centre Hospitalier Lyon-Sud, Hospices Civils de Lyon, 69310 Pierre Bénite, France; 2Laboratoire de Biologie Tissulaire et Ingénierie Thérapeutique, CNRS UMR 5305, 69367 Lyon, Francedominique.sigaudo-roussel@ibcp.fr (D.S.-R.); 3UFR Biosciences, Université Claude Bernard, Villeurbanne, 69100 Lyon, France; 4Service d’Endocrinologie, Diabète et Métabolisme, Groupement Hospitalier Est, Hospices Civils de Lyon, 69677 Bron, France

**Keywords:** diabetic foot, neuropathy, sudomotor function, microcirculation

## Abstract

*Background and Objectives*: According to the International Working Group on Diabetic Foot (IWGDF) risk classification, the estimated risk of developing a diabetic foot ulcer (DFU) is much higher in patients with a history of DFUs (Grade 3) compared to those with a peripheral neuropathy but without a history of DFUs (Grades 1 and 2). It has been suggested that microcirculation impairment is involved in DFU genesis and could be taken into account to refine the existing risk classification. The aim of this study was to evaluate microcirculation parameters in patients with diabetes according to their estimated DFU risk. *Materials and Methods:* A total of 172 patients with type 2 diabetes associated with a peripheral neuropathy and/or a history of DFUs were included and classified into two groups (Grade 1–2 and Grade 3) according to the IWGDF classification. All patients underwent an evaluation of peripheral neuropathy, plantar sudomotor function, and skin microcirculation parameters. These different parameters were compared between both groups. *Results*: There was no significant difference between the two groups in terms of age, diabetes duration, transcutaneous oxygen pressure level, skin microcirculatory reactivity, neuropathy disability score, neuropathy symptom score, or thermal sensitivity. Patients in Grade 3 were more likely to present with retinopathy (OR 3.15, 95%CI [1.53; 6.49]) and severe sudomotor dysfunction (OR 2.73 95%CI [1.29; 5.80] but less likely to have abnormal VPT (OR 0.20 95%CI [0.05; 0.80]). *Conclusions*: The present study found more retinopathy and a more pronounced alteration to sudomotor function in Grade 3 patients, suggesting that these parameters could be considered to better identify patients at high risk of DFUs.

## 1. Introduction

A diabetic foot ulcer (DFU) is a frequent complication of diabetes that can lead to a high burden for patients (hospitalization, amputation, disability, or loss of quality of life) and for society (health resources and cost) [1]. The presence of arteriopathy and bone infection, but also a delay in referral, can increase the rate of hospitalization and amputation [2]. Improved prevention is thus crucial in reducing DFU occurrence. To this end, guidelines were proposed over 20 years ago by the International Working Group on Diabetic Foot (IWGDF) to help detect patients at risk and propose integrated care for DFU prevention [3]. These guidelines are based on a stratified approach, leading to the classification of patients into four groups of different risk levels. Patients without peripheral neuropathy are classified as Grade 0 and present a low risk of DFUs, estimated at 5% at 3 years. Patients with peripheral neuropathy are classified as Grade 1 or Grade 2 if they also present a foot deformity and/or arteriopathy; both grades are associated with a high risk of DFUs of about 15 to 20% at 3 years. Patients with a previous history of DFUs and thus a very high risk of recurrence have an estimated 50% risk at 3 years and are classified as Grade 3 [4]. Despite these guidelines, however, the incidence of DFU events, hospitalization, and amputation remains high in patients with diabetes. This could be partly due to the large risk gap between Grade 1 and 2 and Grade 3 patients [1]. Refining the classification to better detect patients at very high risk of DFUs could help improve the implementation of these guidelines, by better allocating financial and human resources.

In patients with peripheral neuropathy, it has been suggested that microcirculation impairment is involved in DFU genesis and delays in the healing process [5,6,7,8]. These alterations could be detected at an early stage of the disease and guide prevention [9]. Microcirculation, however, has rarely been taken into account when assessing the predictive factors of DFUs but could partly explain the risk gap between Grade 1–2 and Grade 3 patients [10].

The aim of the present study was to compare microcirculation parameters in patients with diabetes classified as either Grade 1 and 2 or Grade 3 according to the IWGDF classification.

## 2. Materials and Methods

### 2.1. Study Population

Patients with type 2 diabetes associated with a peripheral neuropathy and/or a history of DFUs from a single diabetic center between December 2017 and May 2022 were included in a prospective cohort study to assess the predictive factors of DFUs. The study was approved by an ethics committee (CPP nord-ouest III, Lyon, France) and registered in ClinicalTrials.gov (ID: NTC03213093). All participants provided written informed consent prior to participation. Patients with a creatinine level > 300 µmol·L^−1^, respiratory or cardiac decompensation, congenital methemoglobinemia, porphyria, cutaneous lesions of the legs, active DFUs, and those with an allergy or hypersensitivity to local anesthetics were excluded. Patients were then divided into 3 groups according to the IWGDF risk classification: Grade 1 for patients with neuropathy, Grade 2 for patients with neuropathy and/or foot deformity and/or lower limb arteriopathy, and Grade 3 for patients with a history of DFUs.

### 2.2. Data Collection

The following baseline characteristics were collected: age, sex, diabetes duration, history of DFUs and/or amputation, glycemic control, and treatments (antihypertensive drugs, antidiabetic drugs, lipid-lowering drugs, antiplatelet agents, and treatment for painful diabetic neuropathy). Obesity was defined as a body mass index (BMI) above 30 kg/m^2^. Well-controlled diabetes was defined by an HbA1c level below 7%.

Data on microvascular complications (nephropathy and retinopathy) were also collected. Nephropathy was defined by the presence of at least one of the following signs: an estimated glomerular filtration rate below 60 mL/min/1.73 m^2^, positive microalbuminuria, and/or positive proteinuria. Retinopathy was defined by the presence of at least one of the following signs: microaneurysms, retinal hemorrhages, intraretinal microvascular abnormalities, retinal neovascularization, vitreous hemorrhage, maculopathy, and/or a history of laser treatment.

### 2.3. Microcirculation Parameter Evaluation

The measurements of microcirculation parameters were all performed on the lower limbs, in resting patients, in a temperature-controlled room, and are described below.

#### 2.3.1. Assessment of Peripheral Neuropathy

The presence of peripheral neuropathy was determined using a 10 g monofilament 5.07 applied at the plantar level of the great toe, as well as the first and fifth metatarsal heads [3]. Peripheral neuropathy was considered when two out of three answers were incorrect. The neuropathy symptom score (NSS) and neuropathy disability score (NDS) were also evaluated; peripheral neuropathy was defined as NSS ≥ 3 and/or NDS ≥ 6 [11,12]. All patients also underwent thermal discrimination threshold (TDT) and vibration perception threshold (VPT) tests. The TDT was assessed at the dorsal part of the feet using TermoSkin 2.0 (Meteda SRL, San Benedetto del Tronto, Italy). An infrared skin thermometer on the back of the device allowed the measurement of foot temperature. Two hammer faces were present; one was set to the foot temperature measured (used as a reference), while the other was set to various degrees of warmer and colder temperatures. The two hammer faces were placed alternatively on the dorsal part of the foot to determine a warm and a cold TDT. The obtained TDT was compared to the TDT of a reference population without neuropathy [13]. The VPT was assessed on the dorsal side of the great toe using Ultrabiotesiometro (Meteda SRL). Vibratory stimulation was gradually increased from 0 to 35 V at a frequency of 50 Hertz. A VPT above 25 V was considered abnormal [3].

#### 2.3.2. Assessment of Sudomotor Function

Sudomotor function was evaluated using a noninvasive, painless, and easy-to-use device (Sudoscan, Impeto Medical, Paris, France). Patients placed the palms of their hands and the soles of their feet on stainless steel electrodes, and an incremental low direct voltage (<4 V) was applied for about 2 min. Electrochemical skin conductance (ESC), a measure of sudomotor function, was obtained from the ratio between the current that was measured and the voltage applied. A foot ESC under 50 microSiemens (µS) was used to define severe sudomotor dysfunction [14].

#### 2.3.3. Assessment of Skin Microcirculation Parameters

Transcutaneous oxygen pressure (TcPO2) was measured on the dorsal side of the first intermetatarsal space using Radiometer TCM 400 (Perimed France S.A.R.L, Craponne, France). The electrode was heated to 44 °C. The TcPO2 expressed in mmHg was recorded for 20 min.

Skin microvascular reactivity was measured using a laser Doppler probe on the lower portion of the medial surface of the tibia using Periflux 5000 (Perimed, Stockholm Sweden). Cutaneous vasodilation was measured in response to pharmacological, thermal, and mechanical stimulations using acetylcholine (ACh 2%; Sigma, Saint Quentin Fallavier, France), local heating at 42 °C (Peritemp PF4005; Perimed), and locally applied pressure (before and after the topical application of lidocaine, as previously described), respectively [15]. The laser Doppler signals were expressed in arbitrary units (a.u.). All responses were reported as the maximal percent increase in cutaneous vascular conductance from baseline.

### 2.4. Statistical Analysis

Since the estimated risk of DFUs in Grade 1 and Grade 2 patients is similar, these two groups were combined as Grade 1–2 for analysis [4]. In order to confirm that the area under the receiver operating characteristic curve was significantly different from 0.55 with a power of 90%, a risk alpha of 5%, and a prevalence of DFUs of 35%, the required population was estimated to be 160 patients. Continuous variables were described using the mean and standard deviation (SD), and categorical variables were described using numbers and percentages. The two groups (Grade 1–2 and Grade 3) were compared using one-way ANOVA for continuous variables and the Chi-squared test for categorical variables. Variables with significant effect in unadjusted analyses were used in adjusted regression analysis. Statistical significance was set at *p* < 0.05. The results were expressed using odds ratios (ORs) with their 95% confidence interval (95%CI). Analyses were performed with SPSS software (SPSS Statistics V21.0, IBM, Armonk, NY, USA).

## 3. Results

A total of 172 patients were included; the mean age was 68.3 ± 8.3 years old, 73% of the patients were men, and the mean BMI was 30.9 ± 5.3. Diabetes had been present for a mean of 20.6 ± 10.7 years and was treated by insulin in 76% of the cases. Most patients had undergone cardioprotective treatment, notably, statins (75%), angiotensin-converting enzyme inhibitors and/or angiotensin II receptor blockers (77%), and antiplatelet agents (63%). Among the 172 patients, 46 (26.7%) were classified as Grade 1, 55 (32%) as Grade 2, and 71 (41.3%) as Grade 3. The frequency of retinopathy was significantly higher in Grade 3 patients (Table 1).

Regarding the peripheral neuropathy and microcirculation parameters studied, none were significantly different between patients in Grade 1 and those in Grade 2. Compared to patients from Grade 1–2, significantly more patients from Grade 3 had severe sudomotor dysfunction (ESC < 50 µS, 37% vs. 63%, *p* < 0.0001) and abnormal warm and cold TDTs (71% vs. 83%, *p* = 0.035 and 43% vs. 65%, *p* = 0.002, respectively). An abnormal VPT was less frequent in patients from Grade 3 compared to Grade 1–2 (76% vs. 93%, *p* = 0.005). There was no significant difference in skin microvascular reactivity between both groups (Table 2).

In multivariate analysis, patients in Grade 3 were more likely to present with retinopathy (OR 3.15, 95%CI [1.53; 6.49]) and severe sudomotor dysfunction (OR 2.73 95%CI [1.29; 5.80]). Conversely, Grade 3 patients were less likely to have an abnormal VPT compared to Grade 1–2 patients (OR 0.20 95%CI [0.05; 0.80]; Table 3).

## 4. Discussion

The present study found that some microcirculation parameters are more impaired in patients with a history of DFUs (Grade 3) than in patients presenting with peripheral neuropathy but without a history of DFUs (Grades 1 and 2).

Similarly to neuropathy and nephropathy, retinopathy is a known microvascular complication of diabetes and can be a marker of skin microangiopathy [16]. The visual impairment caused by retinopathy could further impact the risk of DFUs by complicating preventive foot care. In the present study, the prevalence of retinopathy was higher in patients with a history of DFUs compared to those with neuropathy but without a history of DFUs, despite the same duration of diabetes in both groups. In line with these findings, a recent systematic review reported that a history of retinopathy was significantly associated with the development of DFUs [17]. Moreover, another more recent retrospective study, which included 522 patients with diabetes followed up for 52 months, found that the presence of retinopathy at baseline, in all grades of the IWGDF classification, was associated with the risk of developing DFUs [18]. Taken together, these results suggest that integrating retinopathy as a predictive factor into each grade of the IWGDF classification may allow us to better identify patients at a higher risk of DFUs.

Regarding peripheral neuropathy, the only neurological parameter that was found herein to differ between Grade 1–2 and Grade 3 patients was the presence of an abnormal VPT. The latter was less frequent in Grade 3 patients, likely due to the criteria used for the IWGDF classification; while all Grade 1–2 patients had a neuropathy, this was not necessarily the case for Grade 3 patients. Nevertheless, the prevalence of an abnormal VPT was high in Grade 3 patients, as expected, given the importance of neuropathy in the pathogenesis of DFUs.

Dryness and cracks due to the impairment of sweat gland function increase skin vulnerability and reduce its ability to repair. Sudomotor function can thus be used as a marker of the skin’s ability to react to local stress (pressure, shear stress, trauma). A more pronounced impairment in sudomotor function was found herein in Grade 3 patients compared to Grade 1–2 patients. Another team has previously shown interest in measuring sudomotor function to predict DFUs [19,20]. In their first cross-sectional study, although sudomotor function assessed by an electromyography apparatus was found to be impaired in patients with an active DFU, the approach used showed high variability and would be difficult to implement in routine practice. In their second study, the authors prospectively assessed the predictive value of a visual indicator test (Neuropad^®^, Harlow, UK) in patients with diabetes but without a history of DFUs and with a very low prevalence of peripheral neuropathy. They reported that an abnormal test was associated with DFU events with a good sensitivity (0.86) and a low specificity (0.46). When the Neuropad^®^ test was performed on both feet in a subset of patients from the present cohort, nearly all patients had an abnormal result; when the results from the Neuropad^®^ test were compared with those of the Sudoscan, only half of the patients with an abnormal Neuropad^®^ test had an alteration according to the Sudoscan. While Neuropad^®^ may thus provide additional information on the risk of DFUs in a population with a low prevalence of peripheral neuropathy, the choice of a quantitative method, such as that proposed by the Sudoscan device, may be more relevant for patients who already have a peripheral neuropathy. Interestingly, Sudoscan measurement has been shown to be associated with diabetic neuropathy in different populations of patients with type 2 diabetes [21,22,23]. Overall, it offers an easy-to-use, painless, fast, and reliable approach to sudomotor function assessment in routine clinical practice [15,24,25].

Skin microcirculation, notably at the foot level, is known to be impaired in patients with diabetes due to both functional and anatomical abnormalities that lead to inadequate local skin perfusion [26,27,28,29,30]. In the present study, there was no significant difference in terms of microvascular reactivity or TcPO2 level between Grade 1–2 and Grade 3 patients. In a previous study, it was found that microvascular reactivity to pressure was impaired in patients with an active DFU compared to patients with no DFUs [9]. The lack of difference observed between the two groups herein, despite using the same measurement method as in previous studies, could be explained by the difference in population.

Overall, while the present findings did not show any difference in skin microvascular reactivity between Grade 1–2 and Grade 3 patients, retinopathy and sudomotor function appear as interesting parameters to refine DFU risk classification. In line with this, some authors have proposed an association between Sudoscan and retinal screening, along with peripheral neuropathy assessment, in patients with diabetes [31], an approach that could easily be implemented in routine care.

The strengths of the present study include its prospective design and the large size of the cohort, which was well characterized regarding microvascular parameters. Moreover, patients in both groups were well matched in terms of age, sex, BMI, and duration of diabetes. To limit variability, all explorations were performed by a single operator for microvascular reactivity tests and another operator for all the other parameters assessed. However, several limitations must be noted. The patients were recruited from a single tertiary diabetic foot center, and no patients with type 1 diabetes were included, preventing the generalizability of the present findings to the larger population of patients with diabetes. The choice not to mix type 1 and type 2 diabetes was made to ensure a better match between patients in terms of age and comorbidities (such as obesity and metabolic syndrome), as these factors are potential confounders regarding microcirculation parameters. Moreover, Grade 1 and 2 patients were combined for analysis as these patients show a similar risk of developing a DFU, but the existence of a difference in microcirculation parameters between these two grades cannot be excluded. A larger cohort would be necessary for such subgroup analyses. Finally, the cross-sectional design prevents any causal relationship from being drawn between factors such as retinopathy, sudomotor dysfunction, and ulcer formation. However, the present results highlight that these factors, which are more prevalent in Grade 3 patients, are good candidates for identifying patients at a higher risk of DFUs.

## 5. Conclusions

The present study found more retinopathy and a more pronounced alteration in sudomotor function in Grade 3 patients, suggesting that these parameters could be considered as better identifying patients at a high risk of DFUs. The prospective follow-up of the present cohort is currently underway in order to assess whether these parameters are associated with an increased risk of DFUs.

## Figures and Tables

**Table 1 medicina-61-00002-t001:** Baseline characteristics of the 172 patients with diabetes included in the study.

	Grade 1–2	Grade 3	*p* Value
N	101	71	
Age (years)	69.2 (8.0)	67.1 (9.0)	0.096
Sex, male	74 (73)	54 (76)	0.409
BMI > 30	60 (59)	33 (47)	0.077
Diabetes duration (years)	19.9 (9.6)	21.9 (11.7)	0.203
HbA1c < 7%	16 (16)	62 (24.5)	0.122
Retinopathy	39 (39)	49 (69)	<0.001
Nephropathy	77 (77)	58 (82)	0.313
Treatment (%)			
• Insulin	82 (81)	50 (70)	0.072
• Antiplatelet therapy	62 (61)	48 (67)	0.250
• ACEis/ARBs	80 (79)	53 (75)	0.301
• Statins	77 (76)	51 (72)	0.316
• Treatment for painful neuropathy	29 (29)	14 (20)	0.122

Data are presented as n (%) or mean ± SD. Abbreviations: BMI, body mass index; ACEis, angiotensin-converting enzyme inhibitors; ARBs, angiotensin II receptor blockers.

**Table 2 medicina-61-00002-t002:** Peripheral neuropathy, sudomotor function, and skin microcirculation assessment results.

	Grade 1–2	Grade 3	*p* Value
N	101	71	
Peripheral neuropathy (%)			
• NSS ≥ 3	78 (77)	54 (76)	0.499
• NDS ≥ 6	86 (85)	60 (85)	0.279
• Abnormal warm TDT	72 (71)	59 (83)	0.035
• Abnormal cold TDT	43 (43)	46 (65)	0.020
• VPT of great toe > 25 V	94 (93)	56 (76)	0.005
Sudomotor function (µS)			
• ESC hand	58 (17.4)	52. (17.1)	0.039
• ESC foot	52.7 (19.9)	38.7 (20.9)	<0.001
• ESC foot < 50 µS	37 (37)	45 (63)	<0.001
Skin microcirculation			
• TcPO2 (mmHg)	55 (12)	55 (12)	0.988
• Skin microvascular reactivity (AU)			
◦ ACh peak	119 (97)	107 (91)	0.271
◦ ACh plateau	7 (23)	8 (22)	0.193
◦ Thermal peak	443 (343)	418 (411)	0.673
◦ Thermal plateau	558 (452)	560 (597)	0.983
◦ Pressure	109 (97)	99 (66)	0.470

Data are presented as n (%) or mean ± SD. Abbreviations: NSS, neuropathy symptom score; NDS, neuropathy disability score; TDT, thermal discrimination threshold; VPT, vibration perception threshold; ESC, electrochemical skin conductance; ACh, acetylcholine.

**Table 3 medicina-61-00002-t003:** Variables associated with a history of diabetic foot ulcers (Grade 3) after multivariate analysis.

Variable	OR [95%CI)]	*p* Value
Retinopathy	3.15 [1.53; 6.49]	0.002
Abnormal warm TDT	1.31 [0.47; 3.70]	0.604
Abnormal cold TDT	2.05 [0.90; 4.64]	0.087
VPT of great toe > 25 V	0.20 [0.05; 0.80]	0.023
ESC foot < 50 µS	2.73 [1.29; 5.80]	0.009

## Data Availability

The datasets generated and/or analyzed during the current study are available from the corresponding author on reasonable request.

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
