# Peer review of "Impairment of Microcirculation Parameters in Patients with a History of Diabetic Foot Ulcers"

_medicina, 2024, doi:10.3390/medicina61010002_

Round 1
Reviewer 1 Report
Comments and Suggestions for Authors
The paper titled "Impairment of microcirculation parameters in patients with a history of diabetic foot ulcer." is carefully read and reviewed. Authors studied microvascular function and sudomotor function in type 2 diabetic patients across different International Working Group on Diabetic Foot (IWGDF) risk grades to identify additional factors contributing to ulcer risk. There are important pitfalls in the paper:
* Study design is inappropriate. Cross sectional design limits the ability to establish causal relationships between factors like retinopathy, sudomotor dysfunction, and ulcer formation.
* Authors included only 172 participants, however, subgroup analyses could benefit from larger cohorts for more definitive conclusions.
* Authors only fucused on type 2 diabetes with peripheral neuropathy, potentially limiting generalizability to other diabetic populations (type 1 DM).
Minor issues:
* Authors must avoid abbreviations in first use.
* check grammar and typos: "...It measurement and the integration of retinopathy could be useful to better assess..." Its measurement...
Author Response
Comments and Suggestions for Authors
The paper titled "Impairment of microcirculation parameters in patients with a history of diabetic foot ulcer." is carefully read and reviewed. Authors studied microvascular function and sudomotor function in type 2 diabetic patients across different International Working Group on Diabetic Foot (IWGDF) risk grades to identify additional factors contributing to ulcer risk. There are important pitfalls in the paper:
1 - Study design is inappropriate. Cross sectional design limits the ability to establish causal relationships between factors like retinopathy, sudomotor dysfunction, and ulcer formation.
We thank the reviewer for this important point. At this stage, we agree that we cannot establish the causal relationships between factors like retinopathy, sudomotor dysfunction, and ulcer formation. However, our results show that these factors, which are more prevalent in Grade 3 patients, are good candidates to identify patients at higher risk of DFU. To confirm these results, the prospective follow-up of the cohort is currently underway.
We have added this point in the Discussion section of the revised version of the manuscript.
2 - Authors included only 172 participants, however, subgroup analyses could benefit from larger cohorts for more definitive conclusions.
We agree with the reviewer that a larger cohort could be useful to perform subgroup analyses and increase the power of the study to detect significant differences in other factors of interest between the two groups. The significant differences observed for sudomotor function, retinopathy, and vibration perception threshold suggest that these factors are good candidates for discriminating between both groups. To our knowledge, our cohort is one of the largest to propose a complete assessment of microcirculation.
3 - Authors only focused on type 2 diabetes with peripheral neuropathy, potentially limiting generalizability to other diabetic populations (type 1 DM).
We have chosen not to mix type 1 and type 2 diabetes to better match patients for age and comorbidities (such as obesity, metabolic syndrome), these factors being potential confounding factors regarding microvascular parameters.
A comment has been added in the Discussion section of the revised version of the manuscript.
Minor issues:
4 - Authors must avoid abbreviations in first use.
We apologize for this mistake, which has been corrected in the revised version of the manuscript.
5 - check grammar and typos: "...It measurement and the integration of retinopathy could be useful to better assess..." Its measurement...
Once again, we apologize for the grammatical errors and typos. The revised version of the manuscript was carefully reviewed for English language by the medical writer of our institution.
Reviewer 2 Report
Comments and Suggestions for Authors
The authors analyzed predictive microcirculation factors for diabetic foot ulcers in patients with type 2 diabetes.
Here are my comments :
- Introdcution is well written, however some sentences are over references ( 5 to 10 references ).
- While the methods are described comprehensively : howevere add a clearer statement of how the methods directly address the primary study objective.
- explain : "renal decompensation"
- Provide a justification for the study size
- Ensure thresholds like NSS ≥ 3 and NDS ≥ 6 is referenced for consistency
- Mention if Sudoscan has been validated in similar populations
- Combining Grades 1 and 2 appears practical : however it might overlap the interpretation of these two stage risks. Add this risk in the discussion
- Were there confounding factors considered in the adjusted regression model ?
Comments on the Quality of English Languageminor editing
Author Response
Comments and Suggestions for Authors
The authors analyzed predictive microcirculation factors for diabetic foot ulcers in patients with type 2 diabetes.
Here are my comments :
1 - Introduction is well written, however some sentences are over references (5 to 10 references).
As suggested, we have modified the paragraph in the Introduction section in order to better cite the references. Moreover, one reference (Flynn MD, Tooke JE. Aetiology of diabetic foot ulceration: a role for the microcirculation? Diabet Med 1992;9(4):320-9.) which dated from 1992, was removed.
2 - While the methods are described comprehensively : howevere add a clearer statement of how the methods directly address the primary study objective.
We agree with the reviewer that the Methods section was not clearly structured. We have thus modified this section to more clearly present the methods used to answer the study objective. More precisely, a paragraph entitled “Data collection” and another entitled “Microcirculation parameter evaluation” have been implemented.
Moreover, the Abstract has been entirely revised to hopefully increase clarity and readability.
3 - explain: "renal decompensation"
We apologize for this misleading terminology. Renal decompensation herein was defined by creatinine level > 300 µmol.l-1. The term has been removed from the revised version of the manuscript.
4 - Provide a justification for the study size
As suggested, the justification of the study size is now described in the statistical analysis paragraph of the Material and Methods section of the revised manuscript.
“In order to detect that area under the receiver operating characteristic curve is significantly different from 0.55 with a power of 90%, a risk alpha of 5% and a prevalence of DFU of 35%, the required population was estimated at 160 patients.”
5 - Ensure thresholds like NSS ≥ 3 and NDS ≥ 6 is referenced for consistency
The choice of these thresholds to define the presence of a peripheral neuropathy was based on the recommendations of an International expert consensus (reference 12, already cited in the original version of the manuscript) according to which the “minimum acceptable criteria for diagnosis of Diabetic Sensory Peripheral Neuropathy were … NDS ≥ 6 with or without NSS ≥ 3.”
Moreover, the NDS ≥ 6 threshold was associated with an increased risk of developing a diabetic foot ulcer in a large cohort of patients with diabetes with and without neuropathy (Abbot CA, Carrington AL, Ashe H, Bath S, Every LS, Griffths J, et al. The North-West Diabetes Foot Care Study: incidence of, and risk factors for, new diabetic foot ulceration in a community based patient cohort. Diabet Med. 2002;10:377–384. doi: 10.1046/j.1464-5491.2002.00698.x.). This reference is now cited in the Methods section of the revised version of the manuscript, and has been added to the reference list.
6 - Mention if Sudoscan has been validated in similar populations
We thank the reviewer for this comment and have added a point along with the appropriate references in the Discussion section of the revised version of the manuscript. The reference list has also been updated accordingly.
“Interestingly, Sudoscan measurement has been shown to be associated with diabetic neuropathy in different populations of patients with type 2 diabetes [21,22,23].”
Angelica Carbajal-Ramírez, Julián A. Hernández-Domínguez, Mario A. Molina-Ayala, María Magdalena Rojas-Uribe, Adolfo Chávez-Negret. Early identification of peripheral neuropathy based on sudomotor dysfunction in Mexican patients with type 2 diabetes. BMC Neurol. 2019; 19: 109. Published online 2019 May 31. doi: 10.1186/s12883-019-1332-4
Jin J, Wang W, Gu T, Chen W, Lu J, Bi Y, Zhu D. The Application of SUDOSCAN for Screening Diabetic Peripheral Neuropathy in Chinese Population. Exp Clin Endocrinol Diabetes. 2018 Sep;126(8):472-477. doi: 10.1055/s-0043-116673. Epub 2017 Sep 11.
Eman Sheshah, Amal Madanat, Fahad Al-Greesheh, Dalal Al-Qaisi, Mohammad Al-Harbi, Reem Aman, Abdul Aziz Al-Ghamdi, Khaled Al-Madani. Electrochemical skin conductance to detect sudomotor dysfunction, peripheral neuropathy and the risk of foot ulceration among Saudi patients with diabetes mellitus. J Diabetes Metab Disord. 2016 Aug 5:15:29. doi: 10.1186/s40200-016-0252-8. eCollection 2015.
7 - Combining Grades 1 and 2 appears practical: however it might overlap the interpretation of these two stage risks. Add this risk in the discussion
We thank the reviewer for raising this important issue. Although different prevention strategies have been recommended by the International Working Group on Diabetic Foot for the two groups, the risk of developing a DFU reported in the literature is quite similar between both groups, i.e. around 15 to 20% at 3 years. Differences between the two groups is mainly related to foot deformities and lower limb arterial disease. We performed subgroup analyses in our cohort (data not shown) and found no significant difference in terms of neuropathy, TcPO2 level, microvascular reactivity, sudomotor function, retinopathy, age, and duration of diabetes between the two groups. However, we cannot exclude a lack of power to detect a difference due to the limited number of patients in both groups (n = 46 for Grade 1, n = 55 for Grade 2).
A comment regarding this point has been added in the Discussion section of the revised version of the manuscript.
“Moreover, Grade 1 and 2 patients were combined for analysis as these patients show a similar risk of developing DFU, but the existence of a difference in microcirculation parameters between these two grades cannot be excluded. A larger cohort would be necessary for such subgroup analyses.”
8 - Were there confounding factors considered in the adjusted regression model ?
Since the two groups were well balanced for the main confounding factors for microcirculation assessment (age, duration of diabetes, metabolic, and cardiovascular treatment) no confounding factor was added to the adjusted regression model.
Comments on the Quality of English Language: minor editing
The manuscript was carefully reviewed for English language by the medical writer of our institution.
Reviewer 3 Report
Comments and Suggestions for Authors
This is an interesting and well conducted study comparing the impairment of microcirculation parameters in patients with diabetes and peripheral neuropathy. Grade 3 patients are those who have a history of foot ulcer.
They more frequently have retinopathy and more impairment of sudomotor function.
The study is very comprehensive and the methods are in the main well described.The introduction, results and discussion sections are well written.
There are two points for attention
Sudomotor function should be described/explained in the introduction and or the methods sections. At present this is left to the last page in the fifth discussion paragraph.
There is no power calculation in the statistical analysis however the sample was large enough to show key differences. Please cover this in the methods and discussion sections.
Author Response
Comments and Suggestions for Authors
This is an interesting and well conducted study comparing the impairment of microcirculation parameters in patients with diabetes and peripheral neuropathy. Grade 3 patients are those who have a history of foot ulcer. They more frequently have retinopathy and more impairment of sudomotor function.
The study is very comprehensive and the methods are in the main well described. The introduction, results and discussion sections are well written.
There are two points for attention
1 - Sudomotor function should be described/explained in the introduction and or the methods sections. At present this is left to the last page in the fifth discussion paragraph.
We apologize as the Methods section was not clearly structured in the original version of the manuscript. We have now modified the presentation of this section, which now contains a paragraph entitled “Data collection” and another entitled “Microcirculation parameter evaluation”. The assessment of sudomotor function is now described in the latter paragraph. We hope that this re-organization of the Methods section will improve the clarity and readability of the paper, notably regarding sudomotor function. Of note, the Abstract has also been entirely revised.
2 - There is no power calculation in the statistical analysis however the sample was large enough to show key differences. Please cover this in the methods and discussion sections.
As suggested, the justification of the study size is now described in the statistical analysis paragraph of the Material and Methods section of the revised manuscript.
“In order to detect that area under the receiver operating characteristic curve is significantly different from 0.55 with a power of 90%, a risk alpha of 5% and a prevalence of DFU of 35%, the required population was estimated at 160 patients.”
Round 2
Reviewer 1 Report
Comments and Suggestions for Authors
Interestingly, the authors seem to have significantly improved and refined the article. While certain issues, such as the retrospective design, persist, many other problems have been resolved. Moreover, the authors’ responses to the points I raised during the previous review were extremely polite and convincing. Therefore, I would like to express my surprise and congratulations. I believe the article should be accepted.